# The Prognostic and Predictive Value of SOX2^+^ Cell Densities in Patients Treated for Colorectal Cancer

**DOI:** 10.3390/cancers12051110

**Published:** 2020-04-29

**Authors:** Tim J. Miller, Melanie J. McCoy, Tracey F. Lee-Pullen, Chidozie C. Anyaegbu, Christine Hemmings, Max K. Bulsara, Cameron F. Platell

**Affiliations:** 1Medical School, University of Western Australia, M507, 35 Stirling Highway, Crawley, WA 6009, Australia; melanie.mccoy@uwa.edu.au (M.J.M.); tracey.lee-pullen@uwa.edu.au (T.F.L.-P.); chidozie.anyaegbu@curtin.edu.au (C.C.A.); cameron.platell@uwa.edu.au (C.F.P.); 2Colorectal Research Unit, St John of God Subiaco Hospital, P.O. Box 14, Subiaco, WA 6904, Australia; 3Curtin Health Innovation Research Institute, Faculty of Health Sciences, Curtin University, Bentley, WA 6102, Australia; 4Department of Anatomic Pathology, Canterbury Health Laboratories, Christchurch 8013, New Zealand; chris.hemmings@cdhb.health.nz; 5Department of Pathology and Biomedical Science, University of Otago Medical School, Christchurch 8013, New Zealand; 6Institute for Health Research, University of Notre Dame, P.O. Box 1225, Fremantle, WA 6959, Australia; max.bulsara@nd.edu.au

**Keywords:** SOX2, colorectal cancer, chemotherapy, prognosis, oxaliplatin, 5-fluorouracil, cancer stem cell

## Abstract

SOX2 (sex-determining region-Y homeobox-2) is a transcription factor essential for the maintenance of pluripotency and is also associated with stem-cell-like properties in preclinical cancer models. Our previous study on a cohort of stage III colon cancer patients demonstrated high SOX2^+^ cell densities were associated with poor prognosis. However, most patients were treated with adjuvant chemotherapy so the prognostic value of SOX2 could not be assessed independently from its value as a predictive marker for non-response to chemotherapy. This study aimed to assess whether SOX2 was a true prognostic marker or a marker for chemotherapy response in a historical cohort of patients, a high proportion of whom were chemotherapy-naïve. SOX2 immunostaining was performed on tissue micro-arrays containing tumor cores from 797 patients with stage II and III colorectal cancer. SOX2^+^ cell densities were then quantified with StrataQuest digital image analysis software. Overall survival was assessed using Kaplan–Meier estimates and Cox regression. It was found that high SOX2^+^ cell densities were not associated with poor overall survival. Furthermore, all patients had a significant improvement in survival after 5-fluorouracil (5-FU) treatment, irrespective of their SOX2^+^ cell density. Therefore, SOX2^+^ cell densities were not associated with prognosis or chemotherapy benefit in this study. This is in contrast to our previous study, in which most patients received oxaliplatin as part of their treatment, in addition to 5-FU. This suggests SOX2 may predict response to oxaliplatin treatment, but not 5-FU.

## 1. Introduction

Colorectal cancer (CRC) is one of the most common cancers worldwide with nearly 1.8 million new cases diagnosed globally in 2018 [1]. The histopathological stage is currently the most reliable prognostic factor for guiding treatment decisions. Most patients with stage I disease can be cured with surgical resection alone whereas patients with stage III disease are also prescribed either neoadjuvant or adjuvant chemotherapy to improve prognosis. Several prognostic markers are utilized for the identification of patients with stage II disease who are at a high risk of relapse and require systemic treatments. However, long-term outcomes after chemotherapy still vary considerably. This indicates that additional biomarkers are required to accurately identify patients who are most likely to benefit from chemotherapy.

Cancer stem-like cells (CSC) are a sub-population of cancer cells with enhanced tumorigenicity, unlimited self-renewal capacity and chemoresistance due to their quiescent nature [2]. SOX2 (sex-determining region-Y homeobox-2) is a transcription factor that is essential during embryonic development and for the maintenance of pluripotent stem cells [3,4,5,6,7,8,9]. Several preclinical studies in a range of cancer models have indicated that SOX2^+^ cancer cells exhibit characteristic CSC properties [10,11,12,13,14,15]. SOX2^+^ cancer cells have been shown to be quiescent, potentially conferring treatment resistance, are able to avoid immune recognition and can also form metastatic niches [11,14]. Furthermore, the combination of these properties with enhanced tumorigenic and self-renewal capacity [10,13,15] suggests that SOX2^+^ cancer cells may contribute to disease progression, potentially having negative impacts on patient prognoses. These studies also collectively indicate that SOX2 may be an accurate marker for the identification of CSC in tumors.

In a previous study, we observed that high SOX2^+^ cell density was an independent prognostic marker for poor survival in patients with stage III colon cancer [16]. However, most of these patients received adjuvant chemotherapy so the true prognostic value of SOX2 could not be independently assessed from its value as a predictive marker for poor response to chemotherapy. This study aimed to assess the prognostic and predictive value of SOX2^+^ cell density in an independent cohort of patients with stage II–III CRC treated between 1990 and 1999. This historic cohort was selected as it contained a higher proportion of chemotherapy-naïve patients, due to the lower rate of adjuvant chemotherapy prescription at the time.

## 2. Results

### 2.1. Patient Cohort and SOX2 Immunostaining

Patient characteristics are shown in Table 1. As might be expected, patients with stage III disease tended to have poor prognostic traits including vascular invasion and perineural invasion, compared to patients with stage II disease. A considerably higher proportion of patients with stage III disease received adjuvant chemotherapy and had poorer five-year survival rates compared to those with stage II disease.

SOX2 staining was predominantly found in the nucleus of cancer cells, consistent with its biological role as a transcription factor (Figure 1). There were 36 patients (8.1%) with stage II and 44 patients (12.5%) with stage III disease considered to have a high SOX2^+^ cell density (Table 1). No significant correlations between high SOX2 expression and clinicopathological variables were found (Appendix A).

### 2.2. Prognosis

Overall survival according to SOX2 expression was assessed in the stage II and stage III cohorts separately. SOX2 was not significantly associated with overall survival in patients with stage II (HR = 0.78; 95%CI 0.47–1.30; *p* = 0.338; Figure 2A) or stage III disease (HR = 1.00; 95%CI 0.69–1.46; *p* = 1.000; Figure 2B). Neither was SOX2 expression associated with cancer-specific survival for stage II (HR = 0.83; 95%CI 0.40–1.69; *p* = 0.600; Figure 2C) nor stage III disease (HR = 0.91; 95%CI 0.59–1.40; *p* = 0.670; Figure 2D).

### 2.3. Survival Benefit from Chemotherapy

The effect of SOX2 on survival benefit from chemotherapy was assessed in patients with stage III disease. There were 132 (37.5%) patients who received chemotherapy and 193 (54.8%) who did not receive any adjuvant chemotherapy treatment. Patients with no record of adjuvant chemotherapy treatment (27; 7.7%) were excluded from this analysis. High SOX2^+^ cell densities were determined using the median value as the cut-off (1.50 cells/mm^2^) for this analysis to allow adequate power for detection of chemotherapy benefit in the SOX2^low^ and SOX2^high^ groups separately (see Table 1). Both groups demonstrated a significant overall survival benefit from chemotherapy, indicating no survival differences exist between groups based on SOX2^+^ cell density (HR = 0.61; 95%CI 0.43–0.87; *p* = 0.006 and HR = 0.52; 95%CI 0.34–0.78; *p* = 0.002, respectively; Figure 3A,B). When using CSS, patients with SOX2^low^ tumors demonstrated no significant survival benefit from chemotherapy (HR = 0.82; 95%CI 0.56–1.22; *p* = 0.329; Figure 3C) whereas those with SOX2^high^ tumors did benefit from adjuvant treatment (HR = 0.59; 95%CI 0.38–0.92; *p* = 0.019; Figure 3D).

## 3. Discussion

The aim of this study was to determine if high tumor SOX2 expression is a prognostic marker and/or predicts survival benefit from chemotherapy in stage II–III CRC. It was hypothesized that patients with a high density of SOX2^+^ cells would have a significantly poorer prognosis and/or would not benefit from chemotherapy. The results did not indicate that SOX2 was a marker for poor prognosis nor that a higher density was associated with poorer response to chemotherapy in this patient cohort. Patients had improved OS following adjuvant chemotherapy, irrespective of the SOX2^+^ cell density in their tumors. With regard to CSS, we found that patients with SOX2^low^ tumors had no benefit from chemotherapy, but those with SOX2^high^ tumors did. The differing results for OS and CSS may be due to misclassification of CRC-related deaths as deaths from other causes or vice versa. This is a common limitation in retrospective cohort studies as the cause of death information was retrieved from the state registry database which relies on accurate reporting of this data.

Our findings are in contrast to our previous study demonstrating that high SOX2 density was significantly associated with poor prognosis in patients with stage III colon cancer [16]. We also previously reported that high SOX2 expression was independently associated with high-grade morphology [16]. This was not observed in this study for patients with stage II nor stage III disease, which may be due to the relatively low proportion of patients reported to have high-grade tumors in this cohort. The different results of the survival analysis may be explained by the inherent differences between the two cohorts, different chemotherapy regimens prescribed or the immunostaining quantification methods used.

A historic cohort (treated from 1990 to 1999) was specifically selected for this study as the lower rate of adjuvant chemotherapy prescription at the time provided an opportunity to assess prognostic value independently from chemotherapy response. However, for those patients who did receive chemotherapy, the regimen also differed from the previous study due to changes to the treatment guidelines over time. In the current study, patients received a combination of 5-fluorouracil (5-FU) and leucovorin post-operatively, whereas patients in the previous study (treated from 2006 to 2010) mostly received 5-FU, leucovorin and oxaliplatin, following demonstration of a survival benefit from the addition of oxaliplatin [17]. This raises the possibility that SOX2 expression may potentially identify tumors that are resistant to oxaliplatin, explaining the contrasting results. This would be consistent with the results of an in-vitro study demonstrating that induced SOX2 expression was associated with resistance to oxaliplatin but not 5-FU in CRC cells [18]. It is, therefore, possible that SOX2 may be a marker of poor response to oxaliplatin-based chemotherapy regimens. However, further investigation will be required to confirm this.

Patients in this study were treated from 1990 to 1999, compared to 2006–2010 in our previous study [16]. Surgical and pathological practices have considerably improved between these periods, likely improving staging, prognostication and outcomes for patients in our previous cohort [16]. For example, variables such as TILs and mucinous histology were reported at a much lower frequency in this study owing to the limited understanding of their prognostic value at the time. However, it should be noted that variables, such as stage, were accurately reported in this study, allowing for the results to be comparable to our previous study.

TMA design and the method of SOX2 quantification were also key differences between the two cohorts, affecting the proportion of patients considered to have high expression. SOX2 is a heterogeneously expressed marker so it is possible that the use of TMAs, particularly where there are not multiple cores available for analysis, may increase the number of false-negative cases thereby decreasing the number of patients considered to have high expression. This is a limitation common to studies utilizing TMAs for analysis. With regards to digital image analysis, Aperio Imagescope Software suite v. 11 was used to quantify SOX2 expression in the previous study and 18% of patients had tumors with a high density of SOX2^+^ cells, compared to 10% of patients in the current study when using StrataQuest for quantification. The use of scattergram-based thresholding in StrataQuest allowed for the removal of false-positive signals (due to background staining or tissue debris, etc.), which likely contributed to the reduction in the proportion of tumors considered to be SOX2^high^ (Appendix A). This was unexpected and the reduced proportion of patients with SOX2^high^ tumors limited our ability to assess the effect of SOX2 density on chemotherapy benefit, hence the use of the median value to stratify patients with stage III disease for this analysis. Importantly, the reduced proportion of patients considered to harbor SOX2^high^ tumors in this cohort is more consistent with what was observed in a previous study of SOX2 in CRC [19]. Revised power calculations with the observed proportion of SOX2^high^ patients in those with stage III disease indicate that the study was still adequately powered (β = 86.3 to detect an HR of 2.0 if α = 0.05) for the assessment of prognostic value. Calculations using the observed proportion of SOX2^high^ patients and the estimated median survival time for those with stage II disease suggest that the prognostic analysis with these patients was underpowered (β = 62.8% to detect an HR of 2.0 if α = 0.05 and median survival of 120 months for SOX2^high^ patients with stage II disease). However, while the effect size was not used in these calculations, there are limitations when using observed variables in post-hoc power calculations [20]. They should, therefore, be interpreted with caution.

The result that SOX2 density was not prognostic in patients with stage II nor stage III disease, while acknowledging that the stage II analysis was likely underpowered, is in contrast to the findings of Lundberg et al. [19] and a meta-analysis by Song et al. [21]. One key difference between these studies is that patients with all stages of CRC (TNM I–IV) were included in the cohort studied by Lundberg et al. [19] and in the majority of cohorts analyzed by Song et al. [21], whereas we assessed the prognostic value of high SOX2 density in stage II and stage III patients separately. The rationale for this is that prognostic markers are used to identify patients at a higher risk of relapse so that they can be offered adjuvant chemotherapy. By performing the analysis in this manner, we focused the prognostic analysis on a group of patients for whom new prognostic markers are required, namely those with stage II disease.

## 4. Materials and Methods

### 4.1. Patients and Ethical Approval

The cohort contained 445 patients with stage II and 352 patients with stage III CRC treated via surgical resection with curative intent with or without adjuvant chemotherapy at Sir Charles Gairdner Hospital (SCGH), Perth, Western Australia, between the years 1990 and 1999. 5-FU/leucovorin-based regimens were prescribed for adjuvant chemotherapy, as per the recommendations at the time. This study and the use of pathological samples and deidentified data were approved by SCGH Human Research Ethics Committee (HREC) (RGS00788) and St John of God Healthcare HREC (SJ-1360) under a waiver of consent.

### 4.2. Tissue Micro-Arrays

Tissue micro-arrays (TMAs) were constructed from formalin-fixed paraffin-embedded tissue as previously described [22]. The TMAs contained one (252 cases), two (516 cases), three (20 cases) or four (9 cases) representative tumor cores per case measuring 1 mm in diameter.

### 4.3. Immunostaining

Sections 4 µm thick were cut from each TMA block and mounted on positively charged slides. Manual immunohistochemistry protocols were then performed for SOX2, as previously described [16]. Deparaffinization was performed in graded xylene and ethanol series before antigen unmasking in 10 mM buffered sodium citrate solution (pH 6.0) for 6 min at 121 °C in a laboratory pressure cooker (DAKO, Glostrup, Denmark). Endogenous peroxidase activity and non-specific IgG interactions were blocked using Peroxidazed 1 and Background Sniper solutions, respectively (Biocare Medical, Pacheco, CA, USA). Sections were then incubated for 1 h at room temperature with an anti-SOX2 rabbit monoclonal antibody solution (1:50, EPR3131, Abcam, Cambridge, UK) diluted in antibody diluent (DAKO, Denmark). A concentration matched isotype control antibody was used as a negative control for all TMA slides. EnVision Rabbit FLEX Link solution was applied to all sections for signal amplification and the REAL EnVision HRP/DAB staining system was used to complete immunostaining, as per the manufacturer’s instructions (DAKO, Denmark). Sections were briefly counterstained in Mayer’s hematoxylin and blued in Scott’s water (Hurst Scientific, Forrestdale, Australia) before dehydration and mounting.

### 4.4. Image Analysis and SOX2 Quantification

Immunostained TMA slides were scanned using the Aperio Scanscope XT high-resolution scanner at 40× magnification (Leica Biosystems, Mt. Waverley, Australia). SOX2 expression was analyzed using StrataQuest v. 6 software (TissueGnostics, Vienna, Austria). Tissue areas were automatically detected by StrataQuest using thresholds on a grayscale image created from the RGB values and reviewed to ensure analyzed tissue areas matched the core histology description (Appendix A). SOX2 expression was then quantified by the software through the detection of DAB^+^ nuclei to count the number of positive cells (Appendix A). Thresholding was performed to ensure SOX2 positive cells were accurately detected for each core (Appendix A). Data were reported in cells/mm^2^.

### 4.5. Statistical Analysis

Cut-offs for high SOX2^+^ cell density were determined by visual review of tissue cores while blinded to all clinical, pathological and outcome information or by use of the median value, as appropriate. The cut-off determined by visual review was 40 cells/mm^2^ and the median value was 1.50 cells/mm^2^. A case was determined as “high” if the mean SOX2 density across all available cores was above the cut-off. Overall survival (OS), defined as time from surgery until death from any cause, was used as the primary endpoint for survival analyses. Cancer-specific survival (CSS), defined as time from surgery until death from colorectal cancer, was used as a secondary endpoint for survival analysis. Kaplan–Meier curves were used to assess survival endpoints and the differences between curves were compared using log-rank estimates. Univariate regression analyses were performed using Cox regression. All results with *p* < 0.05 were deemed significant. The censor date for survival analyses was taken as five years after the surgery date for the last patient in the cohort.

SAS v. 9.4 (SAS Institute, Cary, NC, USA) and GraphPad Prism v. 7 (GraphPad Software, San Diego, CA, USA) were used for statistical analyses.

### 4.6. Power Calculation

Based on our previous study [16], we anticipated 18% of patients with stage III CRC having SOX2-high tumors and median survival time of 66 months in this group. With an accrual time of 90 months and a follow-up time of 60 months, a sample size of 352 patients would give 94.8% power to detect a hazard ratio of 2.0 at α = 0.05.

## 5. Conclusions

This study found no evidence that SOX2 is a significant prognostic marker for poor survival or that benefit from 5-FU-based chemotherapy is affected by SOX2^+^ cell density in this cohort. Further investigation of the prognostic and predictive effects of SOX2 in other cohorts is warranted to evaluate the negative findings of this study. The results of this study contrast to those of our previous study and this may be partially explained by the different chemotherapy regimens that were prescribed for each cohort. Patients in this cohort received 5-FU/leucovorin-based chemotherapy whereas patients in our previous cohort [16] had oxaliplatin added to the regimen. This is an interesting finding and further investigation of SOX2^+^ cell densities and oxaliplatin-based chemotherapy response may be an avenue for future study.

## Figures and Tables

**Figure 1 cancers-12-01110-f001:**
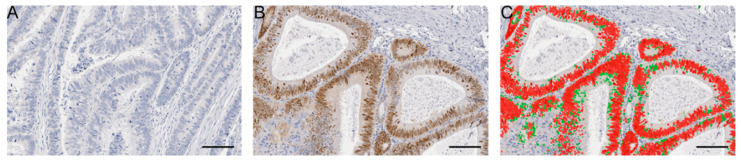
SOX2 (sex-determining region-Y homeobox-2) staining and quantification. Representative images of (**A**) SOX2 low and (**B**) high immunostaining and (**C**) digital analysis of the SOX2 high sample using StrataQuest software. Scale bars 100 μm.

**Figure 2 cancers-12-01110-f002:**
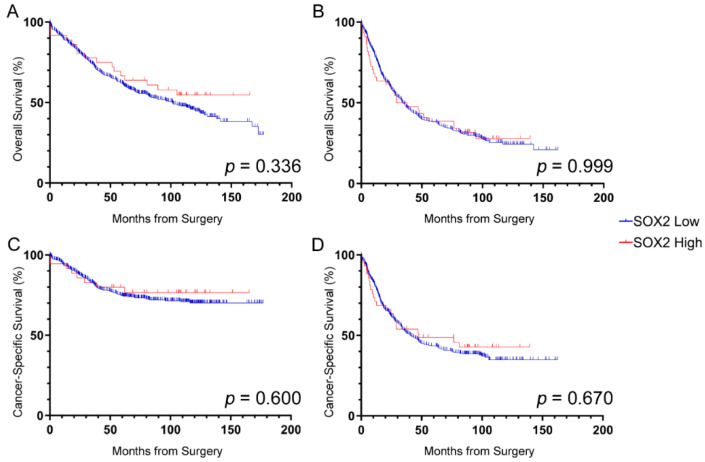
Prognostic value of SOX2^+^ cell densities. Overall survival (**A**,**B**) and cancer-specific survival (**C**,**D**) for patients with (a,c) stage II and (b,d) stage III CRC based on SOX2 expression. No significant associations between SOX2 density and survival were found. Log-rank *p*-values shown.

**Figure 3 cancers-12-01110-f003:**
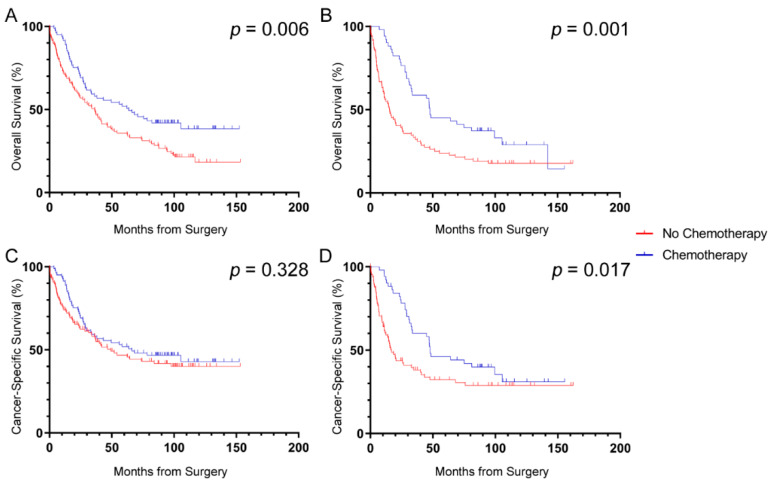
Effect of SOX2 density on survival benefit from chemotherapy in patients with stage III CRC. Patients with stage III disease that had (**A**) low or (**B**) high SOX2 density both demonstrated significant benefit in overall survival from adjuvant chemotherapy. Patients that had (**C**) low SOX2 density did not demonstrate a benefit in cancer-specific survival whereas those with (**D**) high SOX2 density did benefit from chemotherapy. Log-rank *p*-values shown.

**Table 1 cancers-12-01110-t001:** Cohort characteristics by stage.

Variable	AJCC Stage II	AJCC Stage III
	n = 445	n = 352
Age, median (IQR)	72.3 (64.6, 79.0)	68.3 (59.3, 76.9)
T Stage, n (%)		
T1	-	1 (0.3)
T2	-	11 (3.1)
T3	419 (94.2)	331 (94.0)
T4	26 (5.8)	9 (2.6)
Localization, n (%)		
Proximal colon	199 (44.7)	140 (39.8)
Distal colon	117 (26.3)	110 (31.3)
Rectum	113 (25.4)	102 (29.0)
Not Reported	16 (3.6)	0 (0.0)
Vascular Invasion, n (%)		
Present	69 (15.5)	160 (45.5)
Absent	361 (81.1)	192 (54.6)
Not Reported	15 (3.4)	0 (0.0)
PNI, n (%)		
Present	18 (4.0)	39 (11.1)
Absent	407 (91.5)	313 (88.9)
Not Reported	20 (4.5)	0 (0.0)
Grade, n (%)		
High	45 (10.1)	36 (10.2)
Low	395 (88.8)	314 (89.2)
Not Reported	5 (1.1)	2 (0.6)
Mucinous, n (%)		
Yes	72 (16.2)	25 (7.1)
No	74 (16.6)	26 (7.4)
Not Reported	299 (67.2)	301 (85.6)
TILs, n (%)		
Present	58 (13.0)	11 (3.1)
Absent	87 (19.6)	37 (10.5)
Not Reported	300 (67.4)	304 (86.4)
MMR, n (%)		
Deficient	63 (14.2)	21 (6.0)
Proficient	374 (84.0)	323 (91.8)
Not Reported	8 (1.8)	8 (2.3)
SOX2, n (%) ^1^		
Low	409 (91.9)	308 (87.5)
High	36 (8.1)	44 (12.5)
Adjuvant Chemotherapy, n (%)		
Yes	22 (4.9)	132 (37.5)
No	167 (37.5)	193 (54.8)
Not Reported	256 (57.5)	27 (7.7)
5-Year OS, n (%)		
Alive	277 (62.2)	135 (38.4)
Died	168 (37.8)	217 (61.6)
5-Year CSS		
Alive/Died from other cause	343 (77.1)	164 (46.6)
Died from CRC	102 (22.9)	188 (53.4)
Follow-up time, median ^2^	173.7 months	166.1 months

^1^ SOX2^high^ for densities >40 cells/mm^2^ as determined by visual assessment. ^2^ Calculated using reverse Kaplan–Meier method. TILs Tumor-Infiltrating Lymphocytes; PNI Perineural Invasion; MMR Mismatch Repair Status; Met. LNs Metastatic Lymph Nodes; CRC Colorectal Cancer; OS Overall Survival; CSS Cancer-Specific Survival.

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
