# Peer review of "The Prognostic and Predictive Value of SOX2+ Cell Densities in Patients Treated for Colorectal Cancer"

_cancers, 2020, doi:10.3390/cancers12051110_

Round 1

Reviewer 1 Report

There is conflicting evidence in the literature around whether SOX2 expression is associated with survival in colorectal cancer. The authors have provided a follow up study to their 2017 paper, initially reporting that high SOX2 expression was an independent factor for poor cancer-specific survival. In this study, authors have assessed SOX2 expression in stage II and III patients and analysed the association with overall survival and chemotherapeutic response. Whilst the results contradict their earlier study, the authors have proposed that SOX2 may be predictive of response to oxaliplatin, but not 5-FU. This is an interesting concept, although it is not explored in this manuscript. Below are some queries and suggestions that I believe may improve this manuscript.

-The introduction would benefit from some discussion about the oncogenic role of SOX2.

-The authors reported on overall survival and not cancer-specific survival. I believe CSS would be of more interest to your audience. Was CSS assessed? In their 2017 publication, the authors report a >10% difference in survival when comparing numbers for death (overall) to death (cancer-specific). This would be significant in a cohort of this size.

-The authors previously reported that high SOX2 expression was independently associated with high histological grade. Can the authors comment on whether this was observed in the most recent cohort analysis? And indeed, are there associations with other cohort characteristics in this current cohort?

-The authors report that 27 stage III patients have no record of adjuvant chemo. Given this, I believe these patients should be removed from the survival benefit analysis (Figure 3) to ensure the robustness of this analysis.

-Can the authors comment on the treatment regimes received by these patients? It is suggested that these patients received only 5FU and leucovorin. Did they all receive the same treatment/dose? Did all of the patients in this cohort complete the prescribed treatments? Were there inclusion criteria set out patients who underwent chemotherapeutic treatment?

-In their previous study, each case had two central tumour cores as well as one leading edge and one normal epithelium core analysed. In this current study the TMAs contained "at least one representative tumor core for each case." Can the authors comment on how many patients had a single core analysed? Do they believe it is an adequate amount of tissue to accurately assess SOX2 expression?

-The authors acknowledge that the findings in this study are in contrast to their previous study and that of Lundberg et al and they have provided some discussion around this. However, this is also in contrast to a meta-analysis performed by Song et al analysing 13 studies (including Miller et al 2017) but the authors have not addressed this in their discussion. 

Other minor comments

-Can the authors provide some clarity around the tissue samples included in this study i.e. were all of the patients treatment naive at the time of resection; and were the patients considered curative at the time of resection? 

-It would be beneficial to include the cut-offs for SOX2 high and low expression in the methods section and how this was determined.

-Table 1 - Grade and MMR have 'not reported' abbreviated to NR but this is not consistent throughout the table.

-Figure 3 legend, small formatting error.

Reviewer 2 Report

The manuscript is well written and easy to understand. The hypothesis is based on previous experiences and thus fair. The cohorts are of adequate sizes and I appreciate the efforts to publish negative results as well.

  1. The patient cohorts are rather old, resected 20-30 years ago. The authors argue for this. What about the surgical handling and staging. This has certainly improved over an average of 25 years. Are the present cohorts still representative?
  2. Results: One may argue that reporting on differences between TILs in stage II and III may not be realistic as this parameter was not reported for 67-86% of the tumor sections.
  3. Results: In general, it seems as if the rate of “not reported” is more or less the same for stage II and III for all parameters – except for adjuvant chemotherapy. Not reported for stage III in 8% of the cases but in 58% for stage II. They are from the same time period. Why this huge difference?
  4. Table 1: I would prefer to differentiate between right and left colon + rectum and not only proximal and distant as this may be of importance.
  5. Figure 3: even though the “not reported” adjuvant chemotherapy group was only 8% they cannot be added to the adjuvant group as this information is missing. They should be left out. I know it will probably not change the overall message but it is the correct way to do it.
  6. Conclusion: True, we cannot based on the present results rule out an impact on oxaliplatin resistance/benefit. However, if the only prognostic benefit, previously identified, should be attributed to the addition/effect of oxaliplatin this difference would not have been that large considering an average benefit in stage III of 5-7% OS from oxaliplatin.
  7. MM: How is the expression pattern of SOX2 in whole tumor sections and based on that is TMA then validated for this marker or may heterogeneity have influenced the results?
  8. SOX2 quantification: It seems as if the total number (cells/mm2) was used. Have the authors considered some sort of normalization, positive cells / relative to the total number of cells per core?

Reviewer 3 Report

In this research, authors tried to claim the SOX2 high cells' density has no prognostic value in stage II and stage III CRC patients. This research should be meaningful for the clinical prognostic and treatment. However, the experiments and data shown in this manuscript couldn't support the exclusion powerfully. It is necessary to evaluate more cohorts and make the experiments more scientific and convincible. 

Round 2

Reviewer 1 Report

The authors have made satisfactory changes to the manuscript to address the comments and suggestions. No further changes required.

Reviewer 3 Report

The authors address my concern appropriately. I agree to accept this manuscript.